# Cultural Compass: Predicting Transfer Learning Success in Offensive Language Detection with Cultural Features

**Li Zhou**[1][2], **Antonia Karamolegkou**[2], **Wenyu Chen**[1], **Daniel Hershcovich**[2]

[1]University of Electronic Science and Technology of China
[2]Department of Computer Science, University of Copenhagen

li_zhou@std.uestc.edu.cn, antka@di.ku.dk, cwy@uestc.edu.cn, dh@di.ku.dk

## Abstract

The increasing ubiquity of language technology necessitates a shift towards considering cultural diversity in the machine learning realm, particularly for subjective tasks that rely heavily on cultural nuances, such as Offensive Language Detection (OLD). Current understanding underscores that these tasks are substantially influenced by cultural values, however, a notable gap exists in determining if cultural features can accurately predict the success of cross-cultural transfer learning for such subjective tasks. Addressing this, our study delves into the intersection of cultural features and transfer learning effectiveness. The findings reveal that cultural value surveys indeed possess a predictive power for cross-cultural transfer learning success in OLD tasks and that it can be further improved using offensive word distance. Based on these results, we advocate for the integration of cultural information into datasets. Additionally, we recommend leveraging data sources rich in cultural information, such as surveys, to enhance cultural adaptability. Our research signifies a step forward in the quest for more inclusive, culturally sensitive language technologies.[1]

**Warning**: *This paper discusses examples of offensive content. The authors do not support the use of offensive language, nor any of the offensive representations quoted below.*

## 1 Introduction

Cross-lingual transfer has been successful in various NLP tasks (Plank and Agić, 2018; Rahimi et al., 2019; Eskander et al., 2020; Adelani et al., 2022) particularly in scenarios where low-resource languages lack sufficient training data (Lauscher et al., 2020). To ensure successful cross-lingual transfer, leveraging prior knowledge becomes crucial in selecting the most suitable transfer language

and dataset. Most research studies predominantly rely on data-dependent and language-dependent features to predict the optimal language to transfer from (Lin et al., 2019; Sun et al., 2021). However, the same language is often spoken in vastly different cultures, and conversely, similar cultures may speak very different languages. In subjective tasks such as Offensive Language Detection (OLD), relying solely on measures of linguistic proximity and data-specific features for predicting the optimal transfer language/dataset may be insufficient. It is crucial to consider cultural differences.

Cultural sensitivity and adaptation pose a great challenge in NLP tasks (Papadakis et al., 2022; Arango Monnar et al., 2022; Hershcovich et al., 2022; Sittar and Mladenic, 2023). Despite the capabilities of Language Models like ChatGPT, their cultural adaptation to culturally diverse human societies is limited (Cao et al., 2023). In an increasingly globalized and AI-driven world, this gap needs to be addressed to create and sustain an effective communication environment (Aririguzoh, 2022).

Taking steps towards this direction, we focus on the success of cross-cultural transfer learning in OLD. Offensive language can vary greatly depending on cultural backgrounds. While most multilingual OLD datasets are constructed by filtering a predefined list of offensive words (Zampieri et al., 2019; Sigurbergsson and Derczynski, 2020; Jeong et al., 2022; Deng et al., 2022), certain offensive words are culturally specific. For example, OLD models trained on American cultural contexts may struggle to effectively detect offensive words like "m*adarchod" and "pr*sstitute" in Indian texts (Ghosh et al., 2021; Santy et al., 2023). Previous studies (Nozza, 2021; Litvak et al., 2022; Zhou et al., 2023) have highlighted the challenges of cross-lingual transfer in OLD, emphasizing the significance of accurately predicting the optimal language/dataset for transfer before the implementation.

---

[1]Code is available in https://github.com/lizhou21/cultural-compass.

Given this context, we propose six country-level features that quantify the variations in cultural values across different countries. Additionally, we introduce a feature that measures the distance of offensive words between languages. We address three research questions regarding cross-cultural/cross-lingual transfer learning for OLD:

**RQ1.** Can we predict transfer learning success?

**RQ2.** Which types of features (cultural or offensive word distance) are most informative?

**RQ3.** Do the same features contribute to transfer learning success in OLD and other NLP tasks?

Our experiments show the substantial influence of integrating cultural values in predicting language transfer for a culture-loaded task, OLD. We also find that exploring offensive word distance in specific domains benefits the prediction of the optimal transfer datasets. Notably, we observe that the majority of current datasets lack sufficient cultural information, posing a hindrance to the advancement of cross-cultural NLP research. We adopt HARM-CHECK (Kirk et al., 2022) in Appendix A for handling and presenting harmful text in research.

## 2 Related Work

**OLD methods.** There have been many works on offensive language detection. Some works concentrate more on how to improve OLD systems (Goel and Sharma, 2022; McGillivray et al., 2022; Liu et al., 2022). Considering that offensive language online is a worldwide issue and research efforts should not be confined solely to English (Søgaard, 2022), some researchers focus on constructing non-English OLD datasets (Çöltekin, 2020; Sigurbergsson and Derczynski, 2020; Mubarak et al., 2021; Deng et al., 2022; Jeong et al., 2022). Based on these datasets, some works try to apply cross-lingual transfer learning for LOD (Nozza, 2021; Litvak et al., 2022; Arango Monnar et al., 2022; Eronen et al., 2022; Zhou et al., 2023), but performance has so far not taken into account cultural factors. Understanding the cultural nuances and context surrounding offensive language is crucial for developing effective and accurate models for detecting such language.

**Optimal transfer language prediction.** A major challenge in cross-lingual learning is choosing the optimal transfer language. Language proximity is not always the best criterion, since there

are other linguistic properties that could lead to better results such as phonological or syntactic distances (Karamolegkou and Stymne, 2021; Eronen et al., 2022). Other factors that can influence transfer language selection, such as lexical overlap, have shown mixed findings. Some studies report a positive correlation with the cross-lingual model performance (Wu and Dredze, 2019; Patil et al., 2022; de Vries et al., 2022), while others do not support this finding (Pires et al., 2019; Tran and Bisazza, 2019; Conneau et al., 2020b). To automate the process of selecting transfer languages, there have been attempts using a ranking meta-model that predicts the most optimal languages (Lin et al., 2019; Lauscher et al., 2020; Srinivasan et al., 2021; Blazej and Gerasimos, 2021; Srinivasan et al., 2022; Ahuja et al., 2022; Patankar et al., 2022). These mostly rely on data-dependent and language-dependent features, without taking into account cultural background differences during the optimal transfer language prediction process.

**Cultural features.** Recent research has begun to focus on cross-cultural NLP (Hershcovich et al., 2022). Cultural feature augmentation is able to improve the performance of deep learning models on various semantic, syntactic, and psycholinguistic tasks (Ma et al., 2022). Sun et al. (2021) introduce three linguistic features that capture cross-cultural similarities evident in linguistic patterns and quantify distinct aspects of language pragmatics. Building upon these features, they extend the existing research on auxiliary language selection in cross-lingual tasks. However, Lwowski et al. (2022) confirms that offensive language models exhibit geographical variations, even when applied to the same language. This suggests that using language as a proxy for considering cultural differences is overly simplistic. Santy et al. (2023) highlight the impact of researcher positionality, which introduces design biases and subsequently influences the positionality of datasets and models. So in this paper, we consider country information for annotators across various OLD datasets and propose more fine-grained cultural features to enhance the prediction accuracy of transfer learning for cultural-loaded tasks.

## 3 How to predict optimal transfer datasets?

In this section, we formalize the problem as a *transfer dataset ranking* task for cross-cultural/lingual transfer learning. Different from cross-lingual

transfer learning, our experiments involve considering datasets from the same language but with different cultural backgrounds.

Specifically, we give a task $t$ and provide a related dataset set $\mathcal{D} = \{d_1, d_2, \ldots, d_n\}$. Our objective is to develop a ranking method for the low-resource target dataset $d_i^{\text{tgt}} \in \mathcal{D}$, which ranks the other $n - 1$ candidate high re-resource transfer datasets $\mathcal{D}^{\text{tsf}} = \mathcal{D} - d_i^{\text{tgt}} = \{d_1^{\text{tsf}}, d_2^{\text{tsf}}, \ldots, d_{n-1}^{\text{tsf}}\}$. Exhaustive method is the most direct and simple approach, involving training the task model on each candidate transfer dataset and then evaluating it on the target dataset. However, it is time-consuming and resource-wasting.

An effective approach is to utilize the extracted features from the dataset pairs to predict prior knowledge for a cross-cultural task, eliminating the necessity of conducting task-specific experiments.

$$\nu_{d_i^{tgt}, d_j^{tsf}} = \text{extract}\left(d_i^{tgt}, d_j^{tsf}\right).$$

These extracted features encompass various aspects, including statistical properties, linguistic information, and domain-specific characteristics of the datasets. These features are employed to predict a relative score for each transfer dataset pair.

$$r_{d_i^{tgt}, d_j^{tsf}} = \mathcal{R}\left(\nu_{d_i^{tgt}, d_j^{tsf}}; \theta\right),$$

where $\mathcal{R}(\cdot)$ denotes the ranking model, $\theta$ is the parameters of the ranking model.

## 4   What features can be used to predict?

The current features used for transfer learning prediction can be divided into data-dependent and language-dependent categories. Data-dependent features capture statistical characteristics specific to the dataset, while language-dependent features assess language similarities from different perspectives. Appendix B provides further details on these features. However, in cultural-loaded tasks, it is crucial to account for cultural differences. To address this, we propose incorporating six country-level features that quantify variations in cultural values across countries. Furthermore, we introduce a language-level and domain-specific feature that measures the distance between offensive words in language pairs.

### 4.1   Cultural-Dimension Features

The country-level features encompass six cultural dimensions that capture cross-cultural variations in

values, as per Hofstede's theory (Hofstede, 1984), across different countries. Hofstede's Cultural Dimensions Theory serves as a framework utilized to comprehend cultural disparities among countries. A key component of this theory is the Hofstede Value Survey,[2] a questionnaire designed to comparing cultural values and beliefs of similar individuals between cultures. These dimensions are *Power Distance* (*pdi*): reflect the acceptance of unequal power distribution in a society; *Individualism* (*idv*): examine the level of interdependence and self-definition within a society; *Masculinity* (*mas*): explore the emphasis on competition and achievement (Masculine) or caring for others and quality of life (Feminine); *Uncertainty Avoidance* (*uai*): deal with a society's response to ambiguity and efforts to minimize uncertainty, *Long-Term Orientation* (*lto*): describe how societies balance tradition with future readiness; and *Indulgence* (*ivr*): focuse on the control of desires and impulses based on cultural upbringing. These cultural dimensions provide insights into various aspects of a society's values and behaviors, help to understand the cultural variations and preferences within societies and how they shape attitudes, behaviors, and priorities.

We symbolize these features as $C = \{c_i\}$, where $i \in \{pdi, idv, mas, uai, lto, ivr\}$, with each feature value ranging from 0 to 100.[3] To characterize the differences between the transfer dataset and the target dataset in different cultural dimensions, we represent them using the proportional values of each cultural dimension, which is defined as $Rc_i = c_i^{\text{tsf}}/c_i^{\text{tgt}}$. These cultural distances are denoted as *CulDim* $= \{Rc_i\}$.

### 4.2   Domain-Specific Feature

To predict transfer learning success in OLD tasks, we explore using offensive word distance as a feature, inspired by Sun et al. (2021) who introduce Emotion Semantics Distance (ESD) for measuring lexical similarities of emotions across languages. In contrast to their method of calculating distances using 24 emotion concepts (Jackson et al., 2019), we calculate offensive distance based on the alignment of offensive concepts across languages.

Specifically, we utilize Hurtlex, a multilingual lexicon comprising offensive, aggressive, and hate-

---

[2] https://geerthofstede.com/research-and-vsm/vsm-2013/

[3] The cultural dimension values for each country can be found in https://geerthofstede.com/research-and-vsm/dimension-data-matrix/.

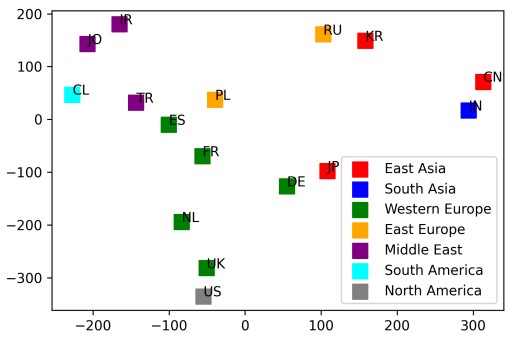

Figure 1: T-SNE Visualization of Cultural Values: Countries represented as data points, with color-coded regions highlighting similarities and differences in cultural dimensions.

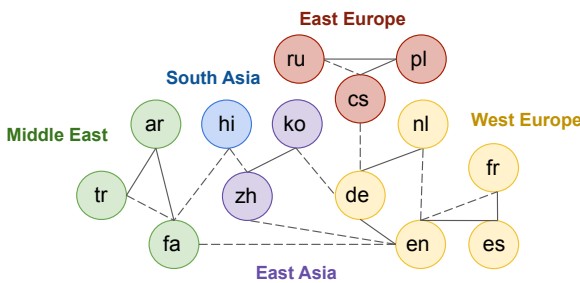

Figure 2: Network based on Offensive Word Distance. Edges connect languages based on their top-2 closest rankings. Solid lines represent mutual ranking, while dashed lines represent exclusive ranking.

ful words over 50 languages ([Bassignana et al., 2018]).[4] This resource offers over 1000 aligned offensive lemmas across languages. For instance, words related to the concept of 'clown' in various languages (i.e. klaun, klown, bouffon, payaso, jokar, Kōmāḷi, 小丑, etc.) are grouped together. We obtain aligned word embeddings based on Fast-Text ([Joulin et al., 2018]),[5] and average the distance between all aligned offensive concepts between language pairs. Mathematically, the offensive distance $OffDist_{(tsf,tgt)}$ from transfer language to target language is defined as follows:

$$OffDist_{(tsf,tgt)} = \sum_{w \in \mathcal{W}} \cos\left(\boldsymbol{v}_w^{tsf}, \boldsymbol{v}_w^{tgt}\right) / |\mathcal{W}|$$

where $\boldsymbol{v}_w^{tsf}$ and $\boldsymbol{v}_w^{tgt}$ respectively represent the aligned word vectors of offensive aligned word $w$ in FastText for transfer language and target language, $\mathcal{W} = \left\{\left(w^{tsf}, w^{tgt}\right)\right\}$ is the set of all offensive aligned words between transfer and target languages. In our example, 'clown' is the offensive word concept belonging to $\mathcal{W}$. Assuming transfer language is English and target language is Czech, then $w^{tsf}$ and $w^{tgt}$ would be the words 'clown' and 'klaun', $\boldsymbol{v}_w^{tgt}$ and $\boldsymbol{v}_w^{tgt}$ denote the corresponding word embeddings extracted with Fasttext respectively.

### 4.3 Features Analysis

To analyze and qualitatively evaluate our proposed features we attempt to visualize them. For *OffDist*

(language-level), we adopt the language selection from [Sun et al. (2021)], including 16 languages. As for *CulDim* (country-level), we select 16 countries based on their relevance to the experimental OLD datasets and the chosen languages.[6]

**Cultural dimensions.** These dimensions provide insights into societal perceptions and interactions across various aspects of life. To visualize the relationships between countries' cultural values, we use T-SNE in Figure [1], where the proximity of points on the plot indicates the degree of similarity in their cultural values. Observations from Figure [1] reveal that there is a degree of consistency between cultural similarity and geographical proximity. For instance, countries in Western Europe and the Middle East tend to be closer in Figure [1]. However, geography alone doesn't solely determine cultural similarity. Notably, the United States and the United Kingdom are visually close despite their geographic distance, while Japan and Germany, though geographically distant, show cultural proximity. This illustrates the intricate interplay between culture and geography, transcending physical boundaries.

**Offensive distance.** Following [Sun et al. (2021)], we create a language network based on offensive distances, shown as Figure [2].[7] Each node corresponds to a language, distinguished by its cultural area. Languages are sorted based on offensive distance, and an edge is drawn between languages that

---

[4]License https://creativecommons.org/licenses/by-nc-sa/4.0/

[5]https://fasttext.cc/docs/en/aligned-vectors.html

[6]The list of selected countries and languages, along with their corresponding abbreviations in Figure [1] and Figure [2], can be found in Appendix [E.1].

[7]Japanese and Tamil are excluded due to lack of aligned FastText embeddings and a Hurtlex lexicon, respectively.

| Dataset | Language | Country | Region | Offensive Label | Source | Size |
|---|---|---|---|---|---|---|
| COLD (Deng et al., 2022) | Chinese | China | East Asia | Offensive | Zhihu, Weibo | 37480 |
| ChileOLD (Arango Monnar et al., 2022) | Spanish | Chile | South America | intentional profanity/vulgarity; unintended profanity/vulgarity; insult/appellation; hate speech | Tweet | 9834 |
| DeTox (Demus et al., 2022) | German | Germany | West Europe | Hate Speech | Tweet | 10278 |
| HinHD (Mandl et al., 2019) | Hindi | India | South Asia | Offensive; Hate Speech; Defamation | Twitter, Facebook, WhatsApp | 8192 |
| KOLD (Jeong et al., 2022) | Korean | South Korea | East Asia | Offensive | NAVER, YouTube | 40429 |
| NJH-UK (Bianchi et al., 2022) | English | United Kingdom | West Europe | Outrage; Insults; Profanity; Character Assassination; | Tweet | 11190 |
| NJH-US (Bianchi et al., 2022) | English | United States | North America | Discrimination; Hostility | Tweet | 9086 |
| PolEval (Ptaszynski et al., 2019) | Polish | Poland | East Europe | Cyberbullying; Hate Speech | Tweet | 10041 |
| TurkishOLD (Çöltekin, 2020) | Turkish | Turkey | Middle East | Offensive | Tweet | 34792 |

Table 1: Statistics of OLD datasets used in our experiments, covering 8 languages and 9 countries.

rank among the top-2 closest. Based on this network we can see that the offensive word distances seem to follow a language proximity pattern. Some clusters seem to appear for the Romance and Slavic languages. 75.8% of the edges correspond to cultural areas, matching exactly the ESD network and surpassing the syntactic network (Sun et al., 2021). Emotion and offensive word similarity therefore seem to have a similar relationship to cultural areas.

**Culture vs. geography.** Based on the above analysis, whether visualizing countries based on *CulDim* or visualizing languages based on *OffDist*, they can approximately fit the grouping based on geographical location but do not strictly match it. This highlights that geographical differences cannot be absolute indicators of cultural distinctions. Culture is a complex and multifaceted concept that is influenced by various factors beyond geography, such as history, politics, migration, and globalization (Taras et al., 2016). Thus, exploring additional culturally relevant features becomes necessary. Appendix C presents the feature correlation analysis involved in the paper.

## 5 Can we predict transfer languages for OLD successfully?

### 5.1 Experimental Settings

**OLD datasets.** To acquire country-level cultural dimension features, we carefully select 9 OLD datasets that include annotators' nationality information, ensuring the availability of country context. The OLD dataset statistics are shown in Table 1 (refer to Appendix D for more details). Offensive language encompasses various aspects (Bianchi et al.,

2022; Demus et al., 2022), including hate speech, insults, threats, and discrimination. In this paper, we simplify the detection tasks in the OLD datasets into binary classification tasks, considering a text offensive if it is labeled with any offensive label.

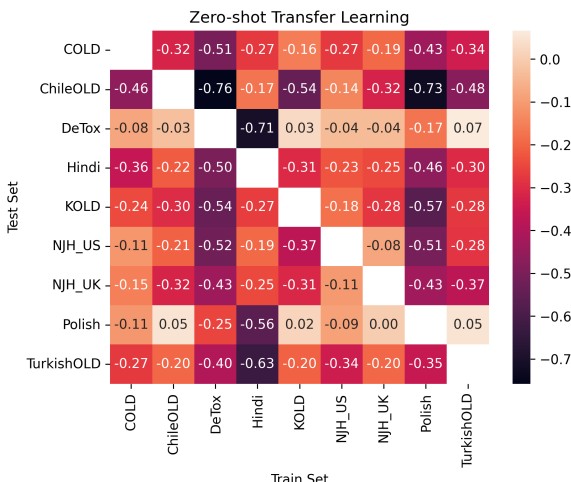

Figure 3: Relative losses (in percent) from zero-shot transfer learning models over the intra-cultural models.

**Zero-shot transfer.** We train intra-cultural XLM-R (Conneau et al., 2020a) for each dataset to ensure cultural consistency. Then, we perform zero-shot transfer (ZST) using XLM-R [8] on 72 directed dataset pairs, simulating cross-cultural implementation. Specifically, for all 72 directed dataset pairs, we fine-tune the XLM-R model on the transfer dataset with learning rate 1e-5. The training epoch is 20, training batch size is 8, we adopt the early-stopping method with patience 10 on the target dataset's dev set. In order to mitigate the influence

---

[8] https://huggingface.co/xlm-roberta-base

| | MAP | NDGC | | MAP | NDGC | | MAP | NDGC |
|---|---|---|---|---|---|---|---|---|
| **LangRank** | 47.48 | 59.33 | **MTVEC** | 51.67 | 68.23 | **Colex2Lang** | 57.78 | 71.41 |
| *+PRAG* | 50.65 | 53.60 | *+PRAG* | 56.98 | 73.87 | *+PRAG* | 55.77 | 71.95 |
| *+OffDist* | 51.08 | 60.17 | *+OffDist* | 64.26 | 76.50 | *+OffDist* | 59.64 | 73.81 |
| *+CulDim* | 51.23 | 60.71 | *+CulDim* | 70.28 | 76.92 | *+CulDim* | 73.25 | 71.56 |
| *+P+O* | 50.90 | 58.35 | *+P+O* | 49.46 | 62.30 | *+P+O* | 55.75 | 73.26 |
| *+P+C* | **57.45** | **67.63** | *+P+C* | 74.44 | 76.53 | *+P+C* | 69.63 | 76.33 |
| *+C+O* | 53.45 | 61.47 | *+C+O* | 73.33 | 73.87 | *+C+O* | **76.69** | **79.34** |
| *+P+C+O* | 56.54 | 62.73 | *+P+C+O* | **76.67** | **79.61** | *+P+C+O* | 68.99 | 72.06 |

Table 2: The performance of transfer language prediction. **Bold** for best performance, underline for second best performance. *P*, *O* and *C* stand for *PRAG*, *OffDist*, and *CulDim* respectively.

| Target Datasets | Ranking comparison | | |
|---|---|---|---|
| **COLD** | KOLD | NJH_US | NJH_UK |
| | TurkishOLD | KOLD | NJH_US |
| **ChileOLD** | NJH_UK | Hindi | NJH_US |
| | KOLD | NJH_UK | COLD |
| **DeTox** | TurkishOLD | KOLD | ChileOLD |
| | KOLD | TurkishOLD | ChileOLD |
| **Hindi** | ChileOLD | NJH_UK | NJH_US |
| | NJH_UK | NJH_US | ChileOLD |
| **KOLD** | NJH_UK | COLD | Hindi |
| | NJH_US | NJH_UK | ChileOLD |
| **NJH_UK** | NJH_US | COLD | Hindi |
| | COLD | NJH_US | Hindi |
| **NJH_US** | NJH_UK | COLD | Hindi |
| | COLD | Hindi | KOLD |
| **PolEval** | ChileOLD | TurkishOLD | KOLD |
| | KOLD | ChileOLD | NJH_UK |
| **TurkishOLD** | ChileOLD | NJH_US | KOLD |
| | ChileOLD | NJH_US | KOLD |

Table 3: Comparison of Top-3 Transfer Datasets. The first line represents ground truth rankings , while the second line represents predicted rankings. Transfer datasets with exact rankings are highlighted in red , and those predicted within the top-3 positions are highlighted in blue .

of randomness in our experiments and ensure result reliability, we run five times for each dataset pair. Subsequently, we compute the average performance (macro F1 score) of the trained model on the target dataset's test set, representing the transfer performance for each dataset pair. Figure 3 illustrates the relative losses incurred by ZST compared to intra-cultural models, highlighting the varying impacts of different transfer datasets on the same target dataset. This presents a challenge in selecting appropriate transfer datasets for low-resource target datasets without prior implementation. These findings underscore the importance of predicting

transfer learning success. Lastly, we obtain the ground truth ranking results for the transfer of OLD based on the experimental results of ZST.

**Transfer ranking prediction.** Following Sun et al. (2021), we use LightGBM (Ke et al., 2017) as the ranking model, known for its effectiveness in ranking tasks. We evaluate the performance using Mean Average Precision (MAP) and Normalized Discounted Cumulative Gain (NDCG) metrics. MAP calculates the average precision across all ranks, while NDCG considers both position and relevance. For our analysis, we focus on the top 3 ranked items for both metrics.

**Baseline features.** We employ the following ranking baselines: LangRank (Lin et al., 2019), incorporating 13 features to rank items, including data-dependent and linguistic-dependent features; MTVEC (Malaviya et al., 2017), learning 512-dimensional language vectors to infer syntactic, phonological, and phonetic inventory features; Colex2Lang (Chen et al., 2023), learning language representations with node embedding algorithms based on a constructed large-scale synset. Additionally, we incorporate *PRAG* features (Sun et al., 2021), capturing cross-cultural linguistic patterns and assessing language pragmatics. Appendix E.3 provides a summary of the features used in each baseline and subsequent feature groups.

## 5.2 Overall Results

We examine the effectiveness of proposed features in predicting transfer dataset rankings for OLD [9]. We specifically incorporate different features and their combinations into the baseline features, and the experimental results are presented in Table 2.

---

[9] The specific cultural dimension features used in our experiments are shown in Appendix E.2.

| | LangRank | | MTVEC | | Colex2Lang | |
|---|---|---|---|---|---|---|
| | MAP | NDGC | MAP | NDGC | MAP | NDGC |
| **BEST** | 57.45 | 67.63 | 76.67 | 79.61 | 76.69 | 79.34 |
| -pdi | 51.52 | 62.46 | 76.28 | 76.31 | 76.34 | 78.95 |
| -idv | 51.98 | 65.18 | 74.81 | 78.52 | 72.06 | 75.86 |
| -mas | 54.09 | 63.20 | 70.94 | 72.00 | 68.92 | 72.74 |
| -uai | 46.12 | 59.55 | 76.76 | 77.89 | 76.06 | 80.96 |
| -lto | 51.19 | 67.07 | 61.94 | 67.16 | 71.80 | 74.36 |
| -ivr | 55.30 | 61.95 | 73.33 | 75.61 | 69.95 | 74.13 |

Table 4: Ablation analysis for the six cultural dimensions, demonstrating that all dimensions contribute to the ranking performance and are therefore meaningful for offensive language detection generally.

Results show that our proposed features, *CulDim* and *OffDist*, independently enhance baseline performance, indicating the influence of cultural-value differences and domain-specific distance on transfer prediction. Moreover, Combining *CulDim* consistently improves ranking prediction, reinforcing the reliability of cross-cultural features in culture-loaded task transfer. Surprisingly, the MTVEV group surpasses the LangRank group, despite lacking data-dependent features. This underscores the impact of culture-dependent and language-dependent features on accurate and robust transfer learning predictions for culture-loaded tasks.

Moreover, using the best results from Table 2, we visualize the top-3 transfer datasets for each target dataset in Table 3. By comparing the ground truth rankings with the predicted rankings, we further assess prediction accuracy. The analysis reveals that for TurkishOLD, the top-3 transfer datasets, including their specific rankings, are accurately predicted. When considering only the prediction accuracy for the top-3 rankings without considering the specific positions, the accuracy rate is 100% for four datasets and 67% for three datasets. Except for NJH_US, the highest-ranked transfer datasets are accurately predicted to be within the top-3 for other datasets. These results demonstrate the potential of our proposed features in reducing trial and error in migration data selection. Accurate prediction of top-3 transfer datasets empowers researchers and practitioners to make informed decisions, enhancing the efficiency and effectiveness of cross-cultural transfer learning.

## 5.3 Cultural Dimensions Analysis

To examine the influence of each cultural dimension on offensive language detection, we perform

an ablation experiment using the optimal feature combination from each baseline setting in Table 2, all of which include *CulDim*. The results in Table 4 demonstrate that almost each cultural dimension can contribute individually to the prediction of OLD transfer. These contributions exhibit variation depending on the specific baseline features utilized. It's important to note that cultural dimensions are not mutually exclusive, and a given culture can exhibit varying degrees of each dimension. Furthermore, the relationship between culture and OLD is complex and can be influenced by factors beyond these dimensions, such as historical context, social norms, and individual differences.

However, the cultural dimension values are somewhat correlated with the sensitivity of offensive language detection. Khan (2014) states that "High power distance employees will be less likely to perceive their supervisory abusive" and "High individualistic cultural employees will be more likely to perceive their supervisory abusive". Bond et al. (1985) find that cultural variations (including Cultural Collectivism and Power Distance) in the use of social control are related to perceptions of an insult and of the insulter. Van Oudenhoven et al. (2008) also highlight that people from different cultures exploit different categories of verbal abuse. Our experimental results further substantiate these research findings.

## 6 What types of features are most important for different types of tasks?

### 6.1 Feature Fine-grained Comparison

To gain a deeper insight into the key features contributing to the success of the optimal transfer dataset prediction in OLD task, we categorize the involved features into eight distinct groups: Data-specific, Topology, Geography, Orthography, Pragmatic, PRAG, OFF, and Cultural. The first five groups align with the grouping proposed by Sun et al. (2021) in their previous work. In particular, the Pragmatic group consists of PRAG features as well as three data-dependent features. So we establish a separate group specifically dedicated to the PRAG features to ensure their distinct recognition and analysis. We introduce two new feature groups: OFF, which incorporates our newly proposed feature *OffDist*, specifically designed for OLD tasks, and Cultural, which encompasses the *CulDim* feature capturing cultural dimensions.

The performance of ranking models trained with

| Datasets | Data-specific | | Typology | | Geography | | Orthography | | Pragmatic | | PRAG | | OFF | | Cultural | |
|---|---|---|---|---|---|---|---|---|---|---|---|---|---|---|---|---|
| | MAP | NDGC | MAP | NDGC | MAP | NDGC | MAP | NDGC | MAP | NDGC | MAP | NDGC | MAP | NDGC | MAP | NDGC |
| COLD | 75.00 | 69.58 | 25.00 | 15.79 | 26.67 | 38.21 | 58.33 | 82.07 | 33.33 | 65.10 | 45.00 | 79.24 | 24.29 | 9.55 | 58.33 | 73.11 |
| ChileOLD | 39.29 | 60.83 | 20.83 | -3.31 | 29.17 | 57.31 | 29.17 | 28.07 | 26.79 | 31.59 | 26.79 | 31.38 | 26.79 | 15.10 | 45.00 | 78.76 |
| DeTox | 22.62 | 31.47 | 26.79 | 47.53 | 24.29 | 35.38 | 33.33 | 59.26 | 26.67 | 27.37 | 26.67 | 22.11 | 66.67 | 77.78 | 62.5 | 76.35 |
| Hindi | 83.33 | 94.34 | 25.00 | 23.59 | 25.00 | 38.21 | 41.67 | 77.00 | 62.50 | 68.41 | 41.67 | 56.35 | 20.83 | 35.02 | 30.95 | 42.69 |
| KOLD | 64.29 | 79.72 | 32.5 | 21.24 | 26.67 | 38.21 | 37.50 | 67.45 | 41.67 | 42.69 | 50.00 | 55.18 | 66.67 | 79.72 | 30.95 | 23.59 |
| NJH_UK | 41.67 | 48.35 | 64.29 | 69.58 | 29.17 | 76.41 | 45.00 | 70.28 | 83.33 | 94.34 | 100.00 | 96.69 | 62.50 | 80.31 | 66.67 | 79.72 |
| NJH_US | 36.67 | 79.72 | 62.5 | 71.93 | 29.17 | 63.68 | 33.33 | 54.59 | 75.00 | 71.93 | 75.00 | 71.93 | 66.67 | 70.17 | 39.29 | 59.66 |
| PolEval | 32.06 | 42.38 | 33.13 | 29.12 | 45.24 | 51.13 | 63.89 | 55.82 | 68.06 | 71.51 | 56.94 | 63.98 | 45.24 | 54.26 | 100.00 | 99.01 |
| TurkishOLD | 26.79 | 52.83 | 70.00 | 67.24 | 40.00 | 53.49 | 24.29 | 24.17 | 66.67 | 79.72 | 32.50 | 65.21 | 22.50 | 55.66 | 50.00 | 46.21 |
| AVG. | 46.86 | 62.14 | 40.00 | 38.08 | 30.6 | 50.23 | 40.72 | 57.63 | **53.78** | 61.41 | 50.51 | 60.23 | 44.68 | 53.06 | 53.74 | **64.34** |

Table 5: Performance comparison of OLD ranking predictions. **Bold** for best performance, underline for second best performance.

different feature groups for OLD is presented in Table 5. The results highlight the varying performance of different feature types. The Cultural group stands out as the best-performing group in both metrics, closely followed by the Pragmatic group. This finding confirms that the proposed *CulDim* feature, which captures cultural-value differences, holds high predictive power for OLD tasks. While the feature *OffDist* can provide some auxiliary predictive power when combined with other features, it does not possess independent prediction ability on its own, which points to the limitations of the domain-specific feature.

| | DEP | | SA | | OLD | |
|---|---|---|---|---|---|---|
| | MAP | NDCG | MAP | NDCG | MAP | NDCG |
| **Data-specific** | 36.94 | 52.83 | 68.00 | 85.40 | 46.86 | 62.14 |
| **Typology** | **58.12** | **79.41** | 49.90 | 60.70 | 40.00 | 38.08 |
| **Geography** | 34.12 | 67.93 | 24.90 | 55.00 | 30.60 | 50.23 |
| **Orthography** | 35.54 | 65.50 | 34.20 | 56.60 | 40.72 | 57.63 |
| **Pragmatic** | 54.37 | 60.50 | **73.20** | **88.00** | **53.78** | 61.41 |
| **PRAG** | 44.45 | 61.11 | 41.81 | 59.58 | 50.51 | 60.23 |
| **Cultural** | 34.47 | 49.23 | 55.80 | 79.75 | 53.74 | **64.34** |

Table 6: Ranking performance comparison on different NLP tasks: dependency parsing (DEP), sentiment analysis (SA) and offensive language detection (OLD). **Bold** for best performance, underline for second best performance.

## 6.2 Comparison in Different Tasks

To quantify the generalization of different feature groups across various NLP tasks, we compare their prediction performance among OLD, Sentiment Analysis (SA) and Dependency Parsing (DEP). Among these tasks, OLD and SA are relatively subjective in nature, while DEP is highly objective, with labels directly derived from linguistic rules. Given that the features of the Cultural group rely on

the country information of the dataset, and the SA and DEP tasks involve datasets that only provide language information (Sun et al., 2021), we filter out the countries where the relevant language is the sole official language.[10] We then calculate the average cultural dimension values of these countries to obtain the cultural dimension value for the language itself.

The comparison results are presented in Table 6. The Topology group demonstrates the highest predictive power for the DEP task. This can be attributed to the inclusion of syntactic distance measurements between language pairs, which aligns with the inherent characteristics of the DEP task. Although the Cultural group does not rank as the most predictive for the SA task, it still achieves significant results. When comparing the performance of the Cultural group across different tasks, it is observed that the performance for OLD and SA tasks is higher than that for the DEP task. This finding highlights the valuable contribution of cultural features to the success of transfer learning predictions for culture-loaded tasks.

## 7 Conclusion

We explored the feasibility of predicting transfer learning success, identifying informative features for transfer learning effectiveness, and examined the generalizability of these features across different NLP tasks. We have gained valuable insights into the potential of incorporating cultural sensitivity in language technologies: dimensions from cultural values surveys consistently improve transfer success prediction, and offensive language distance features provide a further improvement.

We recommend annotation projects to consider

---

[10]List of official languages by country and territory

not only the balance between genres and age groups but also encompass cultures from different regions. Including documentation with information about the cultural background of the annotators enables better interpretation of annotations considering biases and cultural nuances, and promotes a comprehensive and inclusive approach to research and analysis fostering the transparency, usability, and suitability of the dataset. This study highlights the need for culturally informed datasets and the importance of including cultural factors in cross-lingual transfer learning for OLD.

Future work will augment the cross-cultural transfer learning strategy by incorporating cultural features directly into models. Further analysis will also conduct a socially situated multi-cultural human evaluation on language model predictions to address subjective nuances that cannot be identified in reference-based evaluation on isolated sentences. Finally, we will further explore the correlation between features like offensive word distance, geography, and culture, to gain insights into their relationships and effectiveness in capturing offensive language patterns.

## Limitations

While this study provides valuable insights into several aspects of cross-cultural and cross-lingual transfer learning for offensive language detection, there are certain limitations to consider.

The cultural features we introduce certainly do not fully capture the diverse nuances and complexities of different cultural contexts. While more fine-grained than linguistic categories, countries are clearly not homogeneous cultural entities, and assuming they are such can be limiting and harmful to diversity and inclusion. Furthermore, our study is based on a specific set of datasets, tasks, and language pairs, which may limit the generalizability of our findings to other languages and domains. We employ early stopping on the target development set, a practice commonly used in many machine learning settings, while (Kann et al., 2019) state that hand-labeled development sets in the target language are often assumed to be unavailable in the case of zero-shot learning. Moreover, the two most related works (Lin et al., 2019; Sun et al., 2021) do not mention the details about development set.

Regarding the offensive distance feature, it is clear that using embeddings alone for alignment may not capture fine-grained relationships between

languages (Alaux et al., 2019). The coverage of offensive words in many languages is very partial, and furthermore, the ability of aligned word vectors to quantify cultural differences is limited—besides their inability to capture linguistic context, *social* context (who says what to whom, where and when) is crucial to the interpretation of offensive language. This context is very seldom captured in language datasets. This limitation applies in general to the premise of offensive language detection on isolated sentences as a goal and limits the extent to which such classifiers can accurately detect offensive language in the real world.

Despite these limitations, our study serves as an important step toward understanding the role of culture in NLP and provides valuable insights for future research in this domain.

## Ethics Statement

This study involves the use of datasets with potentially harmful and offensive content, but they all follow privacy guidelines to ensure the anonymity and confidentiality of the individuals involved. The datasets used in this study are obtained from publicly available sources, and all efforts have been made to ensure compliance with relevant legal and ethical regulations. We acknowledge the potential impact of offensive language and take precautions to minimize any harm caused. Furthermore, this research aims to contribute to the development of technologies that promote a safer and more inclusive online environment. We are dedicated to fostering a culture of respect, diversity, and fairness in our research practices and encourage open dialogue on the ethical implications of offensive language detection.

## Acknowledgements

Thanks to the anonymous reviewers for their helpful feedback. The authors express their gratitude to Yong Cao, Jiaang Li, and Yiyi Chen for their help during the rebuttal period and to Christoph Demus, Federico Bianchi, Aymé Arango for their assistance with the dataset collection process. Li Zhou acknowledges financial support from China Scholarship Council (No. 202206070002).

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

## A HARMCHECK List

- **Risk of harm protocol**: The specific risks of harm are derived from offensive language detection datasets in multiple languages. Experimenters for this work have access to these datasets, but they don't need to read them carefully.

- **Preview**: The work includes a Content Warning directly after Abstract. The content warning is in italics to maximize visibility.

- **Distance**: Only a few offensive words are included to explain cross-cultural differences in the introduction section.

- **Respect**: For slurs and profanities in text, the authors star out the first vowel with an asterisk ("*").

## B Data-dependent and Language-dependent Features

The existing features used to predict transfer learning are mainly divided into data-dependent and language-dependent features.

### B.1 Data-dependent Features

Data-dependent features primarily manifest in three aspects: dataset size, Type-Token Ratio (TTR), and word overlap, which are specific measurements obtained from the datasets themselves.

**Dataset size.** *Transfer Size* and *Target Size*, serving as fundamental features, indicate the number of data entries utilized for training in transfer and target datasets, denoted by $\mathcal{S}_1$ and $\mathcal{S}_2$ respectively. Additionally, the *Ratio Size* $\mathcal{S}_r = \frac{\mathcal{S}_1}{\mathcal{S}_2}$ is utilized to indicate the relative size difference between the transfer dataset and the target dataset. It quantifies how much larger the transfer dataset is in comparison to the target dataset.

**Type-Token Ratio** TTR is the ratio between the number of types (vocabulary set) and the number of tokens in a dataset, measuring lexical variation. A higher TTR represents a higher lexical variation. We utilize *Transfer TTR* $\text{TTR}_1$ and *Target TTR* $\text{TTR}_2$ to represent the Type-Token Ratio (TTR) of the transfer and target datasets, respectively. The lexical-variation *Distance TTR* between the transfer and target datasets is defined as $\text{TTR}_d = \left(1 - \frac{\text{TTR}_1}{\text{TTR}_2}\right)^2$.

**Word Overlap** *Word overlap* is used to measure the lexical similarity between a pair of languages (datasets). It is defined as $\frac{|V_1 \cap V_2|}{|V_1| + |V_2|}$, where $V_1$ and $V_2$ represent the vocabularies of transfer dataset and target dataset.

### B.2 Language-dependent Features

Language-dependent features encompass six linguistic distances queried from the URIEL Typological Database (Littell et al., 2017), as well as three linguistic features (PRAG) that manifest in linguistic patterns and quantify distinct aspects of language pragmatics (Sun et al., 2021).

**URIEL.** The feature vector in URIEL encompasses various linguistic features that describe the typological properties of languages, which include *Genetic*, *Syntactic*, *Phonological*, *Inventory*, *Geographic* and *Featural*. The derived distances quantify the similarities or dissimilarities between languages based on these features.

- *Genetic:* The Genetic distance derived from the Glottolog tree (Hammarstrm et al., 2015) of language families, where it quantifies the dissimilarity between two languages by measuring their distance within the tree.

- *Syntactic:* The syntactic distance is the cosine distance between syntax features of language pairs. The syntax features are adapted from the World Atlas of Language Structures (WALS) (Dryer and Haspelmath, 2013), Syntactic Structures of World Languages (SSWL) (Collins and Kayne, 2011) and Ethnologue [11].

- *Phonological:* The phonological distance measure the cosine distance between vectors containing phonological information from Ethnologue and WALS.

- *Inventory:* The inventory distance is the cosine distance between the inventory feature vectors of languages, sourced from the PHOIBLE database (Moran et al., 2014).

- *Geographical:* Geographical distance can represent the shortest distance between two languages on the surface of Earth's sphere. It is another component of URIEL, in which the geographical vectors of each language express geographical location with a fixed number of dimensions and each dimension representing the same feature.

- *Featural:* The cosine distance between vectors incorporating features from the above five.

**PRAG.** These features are pragmatically-inspired linguistic features that capture cross-cultural similarities manifested in linguistic patterns.

- *Language Context-level Ratio* (*LCR*) measures the extent to which the language leaves the identity of entities and predicates to context. It only takes into account noun probabilities and verb probabilities here, which represent the likelihood of nouns and verbs occurring in the text, respectively.

---

[11] https://www.ethnologue.com/

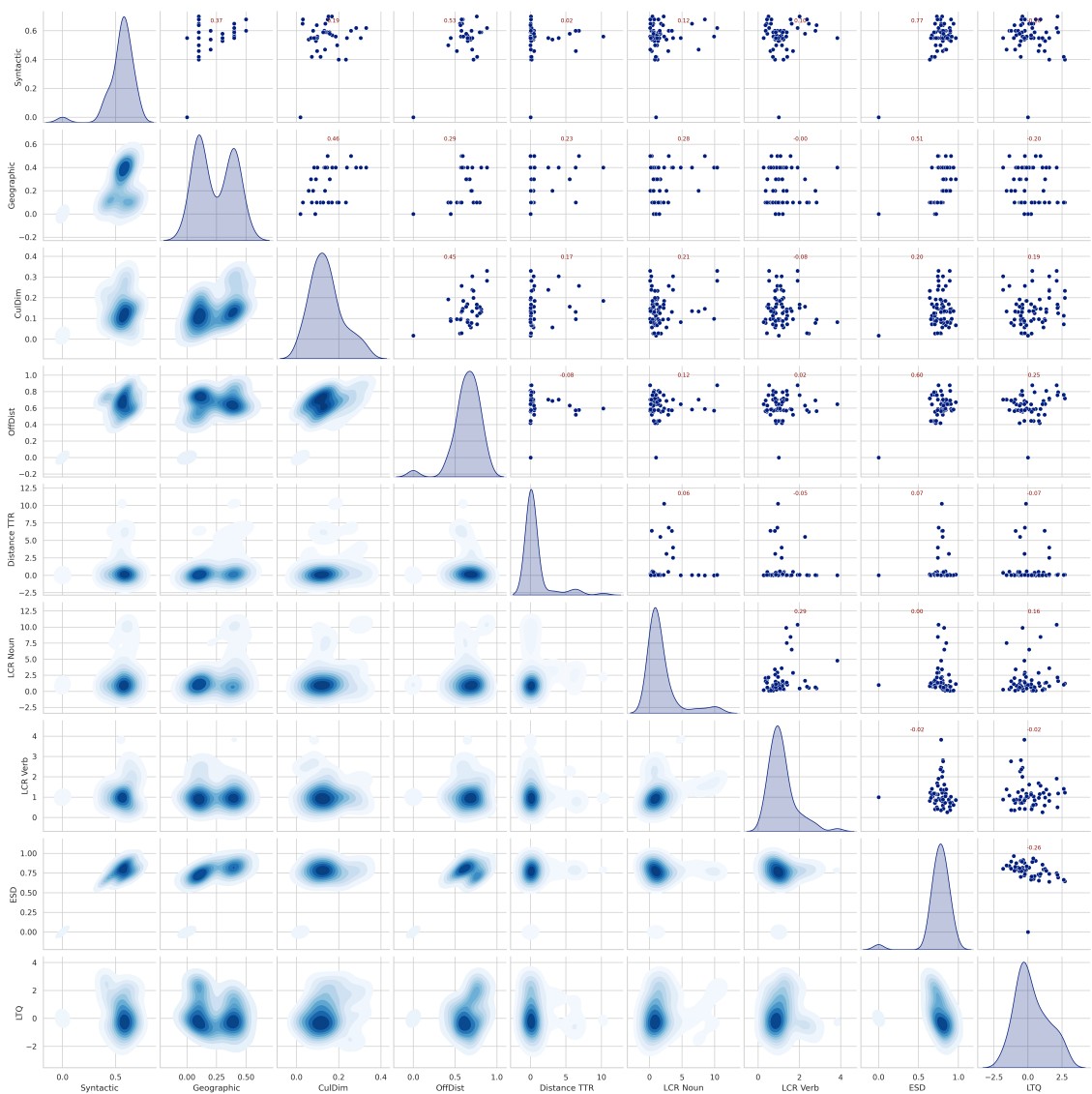

Figure 4: A pairplot showcasing the relationship between culturally relevant features using kernel density estimation (KDE), accompanied by correlation coefficients (in red digit) annotated on the corresponding scatter plots. The diagonal plots display the distribution of each feature using KDE curves, while the lower triangle depicts the joint distributions and contour plots using a color gradient (Blues colormap).

- *Literal Translation Quality* (*LTQ*) quantifies how well a given language pair's MWEs are preserved in literal (word-by-word) translation, using a bilingual dictionary.

- *Emotion Semantics Distance* (*EDS*) measures how similarly emotions are lexicalized across languages.

## C   Feature correlation analysis

### C.1   Different distances

In this section, we discuss the correlations between our different features, with a specific focus on data-independent features. Among all the features, we

have selected those that are associated with cultural diversity, such as *CulDim*, *Geographical*, *Syntactic*, and so on. We construct a sample set by sampling from the OLD zero-shot experiment, which involves all possible pairs of data for the relevant features. Figure 4 shows the relationship between the selected features. As depicted in Figure 4, we observe a strong positive correlation between *CulDim* and *Geographical*, suggesting that there is a relationship between cultural differences and geographical differences to some extent. The correlations between different features vary significantly, highlighting the importance of introducing multidimensional features to a certain extent.

## C.2 Cultural dimension

We conduct a fine-grained correlation analysis for each dimension feature in the *CulDim*. As shown in the Figure 5, *Indulgence (ivr)* and *Individualism (idv)* exhibit a relatively strong positive correlation, implying that societies or groups that tend to score high on indulgence also tend to score high on individualism. This could indicate a cultural inclination towards personal freedom, self-expression, and a focus on individual rights and achievements. However, *Power Distance (pdi)* and has strong negative correlation with *Indulgence (ivr)* and*Individualism (idv)*. This indicates that societies or groups characterized by a high power distance, where hierarchical structures and authority are emphasized, tend to have lower levels of indulgence and individualism. It suggests a cultural tendency towards obedience, respect for authority, and collective orientation over personal freedom and self-expression.

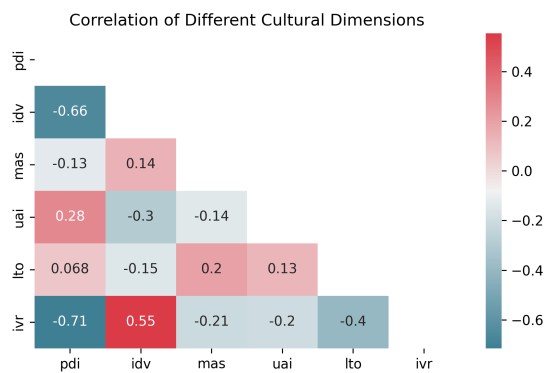

Figure 5: Fine-grained correlation analysis of dimension features in *CulDim*.

## D Datasets details

**COLD** (Deng et al., 2022) is a Chinese dataset, covering the topics of racial, gender, and regional bias. It is collected by strategies of keyword querying and related sub-topics crawling, and labeld by 17 native Chinese native workers.

**ChileOLD** (Arango Monnar et al., 2022) contains tweets in the Spanish language that originated in Chile, which is a representative of the Spanish spoken in South America. Each tweet is annotated with several fine-grained offensive categories by three native Chileans.

**DeTox** (Demus et al., 2022) is a German dataset that is far more comprehensive with twelve different annotation categories. The annotation schema is developed with first-hand users from a reporting office for offensive comments in Germany. The dataset is collected by manually creating keyword list with the help of Google Trends.

**HinHD** (Mandl et al., 2019), utilized in the HASOC 2019 shared task, was collected from Twitter using hashtags and keywords associated with offensive content. In the labeling process, multiple junior annotators participated via an online system to evaluate the tweets.

**KOLD** (Jeong et al., 2022) is a Korean offensive language dataset comprising comments from NAVER news and YouTube platform, which is also filtered by using predefined keywords.

**NJH** (Bianchi et al., 2022) is a English dataset collected with keywords and associated with immigration in the US and/or UK. It is annotated by ten undergraduate researchers from universities in the UK and US, based on which the dataset can be split into *NJH-UK* and *NJH-US*.

**PolEval** (Ptaszynski et al., 2019) is the first dataset for the Polish language containing annotations of harmful and toxic language. The dataset was automatically collected from Twitter accounts in the Polish language and annotated by layperson volunteers under the supervision of a cyberbullying and hate-speech expert.

**TurkishOLD** (Çöltekin, 2020) is the first corpus of offensive language for Turkish and consists of randomly sampled micro-blog posts from Twitter. The annotators are native speakers of Turkish, and all are highly educated.

## E Table Summary for Supplementary Information

By presenting this information in a table format, it allows for easy reference and understanding of the selection criteria and categorization of the data. This section serves to enhance the clarity and comprehensiveness of the paper, ensuring that readers have access to all the necessary details regarding the countries, languages, and their corresponding abbreviations used in the study.

### E.1 Overview of Countries, Languages, and Groups in Feature Analysis

To provide a concise overview the countries and languages included in the feature analysis, we provides a detailed Table 7 showcasing the abbreviations used to represent each country and language in the feature analysis figures (Figure 1 and Figure 2), as well as their corresponding groups based

on relevant criteria.

| | | Country | | Language |
|---|---|---|---|---|
| **East Asia** | CN | China | zho | Chinese |
| | JP | Japan | jpn | Japanese |
| | KR | South Korea | kor | Korean |
| **South Asia** | IN | India | hin | Hindi |
| | | | tam | Tamil |
| **Western Europe** | NL | Netherlands | nld | Dutch |
| | FR | France | fra | France |
| | UK | the United Kingdom | eng | English |
| | ES | Spain | spa | Spanish |
| | DE | German | deu | German |
| **East Europe** | PL | Poland | pol | Polish |
| | RU | Russia | rus | Russian |
| | | | ces | Czech |
| **Middle East** | TR | Turkey | tur | Turkish |
| | JO | Jordan | ara | Arabic |
| | IR | Iran | fas | Persian |
| **South America** | CL | Chile | | |
| **North America** | US | the United States | | |

Table 7: Overview of Countries, Languages, and Groups in Feature Analysis

## E.2 Cultural Dimension Values of Countries and Languages

We display the cultural dimension values associated with the countries and languages involved in our experiments. In OLD, the cultural dimension values are depicted in Figure 8. The datasets used in OLD contain country information, allowing for the direct utilization of corresponding cultural dimension values. However, in the case of SA and DEP tasks, where only language information is provided in the datasets, direct indexing of cultural dimension values is not feasible. As a solution, we select countries where each language is considered the sole official language and employ the average cultural dimension values of these countries to represent the cultural features associated with the respective language. The value details are shown in Table 9.

## E.3 Feature List of Baselines and Feature Groups

Table 10 provides a comprehensive list of features utilized in the baselines and feature groups employed for comparison in the study. It showcases specific sets together for analysis and evaluation purposes.

| Country | pdi | idv | mas | uai | lto | ivr |
|---|---|---|---|---|---|---|
| China | 80 | 20 | 66 | 30 | 87 | 24 |
| Chile | 63 | 23 | 28 | 86 | 31 | 68 |
| Germany | 35 | 67 | 66 | 65 | 83 | 40 |
| India | 77 | 48 | 56 | 40 | 51 | 26 |
| South Korea | 60 | 18 | 39 | 85 | 100 | 29 |
| United Kingdom | 35 | 89 | 66 | 35 | 51 | 69 |
| United States | 40 | 91 | 62 | 46 | 26 | 68 |
| Poland | 68 | 60 | 64 | 93 | 38 | 29 |
| Turkey | 66 | 37 | 45 | 85 | 46 | 49 |

Table 8: Cultural-value features in OLD.

| Language | Country | Cultural Value | | | | | |
|---|---|---|---|---|---|---|---|
| | | pdi | idv | mas | uai | lto | ivr |
| Arabic | Egypt | 80 | 37 | 55 | 55 | 42 | 0 |
| | Jordan | 70 | 30 | 45 | 65 | 16 | 43 |
| | Kuwait | 90 | 25 | 40 | 80 | -1 | -1 |
| | Lebanon | 62 | 43 | 48 | 57 | 22 | 10 |
| | Libya | 100 | 35 | 66 | 67 | 15 | 74 |
| | Qatar | 93 | 25 | 55 | 80 | -1 | -1 |
| | Saudi Arabia | 72 | 48 | 43 | 64 | 27 | 14 |
| | Syria | 80 | 35 | 52 | 60 | 30 | -1 |
| | Tunisia | 70 | 40 | 40 | 75 | -1 | -1 |
| | United Arab Emirates | 74 | 36 | 52 | 66 | 22 | 22 |
| Chinese | China | 80 | 20 | 66 | 30 | 87 | 24 |
| Czech | Czech Republic | 57 | 58 | 57 | 74 | 70 | 29 |
| Dutch | Netherlands | 38 | 80 | 14 | 53 | 67 | 68 |
| | Suriname | 85 | 47 | 37 | 92 | -1 | -1 |
| English | Australia | 38 | 90 | 61 | 51 | 21 | 71 |
| | Ghana | 80 | 15 | 40 | 65 | 4 | 72 |
| | Jamaica | 45 | 39 | 68 | 13 | -1 | -1 |
| | Namibia | 65 | 30 | 40 | 45 | 35 | -1 |
| | Nigeria | 80 | 30 | 60 | 55 | 13 | 84 |
| | Sierra Leone | 70 | 20 | 40 | 50 | -1 | -1 |
| | Trinidad and Tobago | 47 | 16 | 58 | 55 | 13 | 80 |
| | United Kingdom | 35 | 89 | 66 | 35 | 51 | 69 |
| | United States | 40 | 91 | 62 | 46 | 26 | 68 |
| | Zambia | 60 | 35 | 40 | 50 | 30 | 42 |
| France | Burkina Faso | 70 | 15 | 50 | 55 | 27 | 18 |
| | France | 68 | 71 | 43 | 86 | 63 | 48 |
| | Senegal | 70 | 25 | 45 | 55 | 25 | -1 |
| German | Austria | 11 | 55 | 79 | 70 | 60 | 63 |
| | Germany | 35 | 67 | 66 | 65 | 83 | 40 |
| Hindi | India | 77 | 48 | 56 | 40 | 51 | 26 |
| Japanese | Japan | 54 | 46 | 95 | 92 | 88 | 42 |
| Korean | South Korea | 60 | 18 | 39 | 85 | 100 | 29 |
| Persian | Iran | 58 | 41 | 43 | 59 | 14 | 40 |
| Polish | Poland | 68 | 60 | 64 | 93 | 38 | 29 |
| Russian | Russia | 93 | 39 | 36 | 95 | 81 | 20 |
| Spanish | Argentina | 49 | 46 | 56 | 86 | 20 | 62 |
| | Chile | 63 | 23 | 28 | 86 | 31 | 68 |
| | Colombia | 67 | 13 | 64 | 80 | 13 | 83 |
| | Costa Rica | 35 | 15 | 21 | 86 | -1 | -1 |
| | Dominican Republic | 65 | 30 | 65 | 45 | 13 | 54 |
| | El Salvador | 66 | 19 | 40 | 94 | 20 | 89 |
| | Guatemala | 95 | 6 | 37 | 98 | -1 | -1 |
| | Honduras | 80 | 20 | 40 | 50 | -1 | -1 |
| | Mexico | 81 | 30 | 69 | 82 | 24 | 97 |
| | Panama | 95 | 11 | 44 | 86 | -1 | -1 |
| | Spain | 57 | 51 | 42 | 86 | 48 | 44 |
| | Uruguay | 61 | 36 | 38 | 98 | 26 | 53 |
| Tamil | Sri Lanka | 80 | 35 | 10 | 45 | 45 | -1 |
| Turkish | Turkey | 66 | 37 | 45 | 85 | 46 | 49 |

Table 9: Cultural Dimension Values of Languages in SA and DEP Tasks. In particular, When the dimension score is −1, it indicates that the country does not have a score in that specific cultural dimension.

| Type | Feature | Baselines | | Feature Group | | | | | Ours | Other Group |
|---|---|---|---|---|---|---|---|---|---|---|
| | | LangRank | MTVEC | Data-specific | Topology | Geography | Orthography | Pragmatic | Cultural-value | PRAG |
| Data-dependent | Transfer Size | ✓ | | ✓ | | | | | | |
| | Target Size | ✓ | | ✓ | | | | | | |
| | Ratio Size | ✓ | | ✓ | | | | | | |
| | Transfer TTR | ✓ | | | | | | ✓ | | |
| | Target TTR | ✓ | | | | | | ✓ | | |
| | Distance TTR | ✓ | | | | | | ✓ | | |
| | Word overlap | ✓ | | | | | ✓ | | | |
| Data-independent | Genetic | ✓ | | | ✓ | | | | | |
| | Syntactic | ✓ | | | ✓ | | | | | |
| | Featural | ✓ | | | ✓ | | | | | |
| | Phonological | ✓ | | | ✓ | | | | | |
| | Inventory | ✓ | | | ✓ | | | | | |
| | Geographic | ✓ | | | | ✓ | | | | |
| | LCR$_{verb}$ Distance | | | | | | | ✓ | | ✓ |
| | LCR$_{noun}$ Distance | | | | | | | ✓ | | ✓ |
| | ESD Distance | | | | | | | ✓ | | ✓ |
| | LTQ score | | | | | | | ✓ | | ✓ |
| | Rep_Diff | | ✓ | | | | | | | |
| | Power Distance (pdi) | | | | | | | | ✓ | |
| | Individualism (idv) | | | | | | | | ✓ | |
| | Masculinity (mas) | | | | | | | | ✓ | |
| | Uncertainty Avoidance (uai) | | | | | | | | ✓ | |
| | Long-Term Orientation (lto) | | | | | | | | ✓ | |
| | Indulgence (ivr) | | | | | | | | ✓ | |

Table 10: Feature Groups and Baseline Features in the Study