# OpenReview forum: "Cultural Compass: Predicting Transfer Learning Success in Offensive Language Detection with Cultural Features"
_EMNLP/2023/Conference — EMNLP 2023 Findings_

### Official Review · Reviewer_cZgm · 2023-07-31

**Soundness:** 3

**Excitement:**

2: Mediocre: This paper makes marginal contributions (vs non-contemporaneous work), so I would rather not see it in the conference.

**Paper Topic And Main Contributions:**

This paper proposes to incorporate cultural differences for better language transfer, especially for offensive language detection task.
Previous studies have mainly focused on linguistic difference, but this paper firstly investigate why and how to use the cultural difference for language transfer.
Specifically, country information is used as a proxy for the cultural features in this paper, and existing lexicon is employed to compute offensive distance.

**Questions For The Authors:**

- I see that the authors are gonna perform experiments with pre-trained language models in future, but I'm curious why the authors did not do it in this paper. There are few small-sized language models (e.g., BERT-small,base), so it is possible to do experiments with such small pre-trained language models. The language models are known to have knowledge of various aspects (e.g., linguistic, cultural, commonsense, etc.), so it is necessary to show that the proposed features are valuable even if we use pre-trained language models.

**Reasons To Accept:**

- Well organized paper with interesting motivation.
- New features for cultural difference and offensive word distance are proposed, and its effectiveness is demonstrated by experimental results.

**Reasons To Reject:**

- Limited interpretation of the results of Table 3. I see that the proposed 'cultural features' gave much performance gain, and authors insist that the cultural features hold strong predictive power for the OLD task. However, for some datasets of Asian languages (e.g., KOLD, Hindi), the cultural features quite lower power than many other feature groups. The experiments are performed on limited languages, and some languages give contradictory experimental results, thus I think the authors can't insist that the cultural features have 'strong' power than others.
- More details of the zero-shot transfer (ZST) process is required. How the authors 'fine-tune' the model on the transfer dataset? (e.g., training objectives, loss, and etc.)
- The ZST is employed to generate ground-truth ranking results. This is based on very strong assumption that the ZST is perfect and always gives correct answers. Authors need to justfy this.

**Reproducibility:**

3: Could reproduce the results with some difficulty. The settings of parameters are underspecified or subjectively determined; the training/evaluation data are not widely available.

**Reviewer Confidence:**

3: Pretty sure, but there's a chance I missed something. Although I have a good feel for this area in general, I did not carefully check the paper's details, e.g., the math, experimental design, or novelty.

---

> ### Author Rebuttal · Authors · 2023-08-28
>
> Thanks for your meticulous review. We summarize your comments and organize our response in the following four aspects, addressing the rejected reasons and the raised questions. We hope that our replies will make you consider adjusting your score.
>
> **R1. Questioning the ability of cultural features, especially for some datasets of Asian languages (e.g., KOLD, Hindi).**
>
> A1. It needs to be clarified that the predictive challenge we aim to address is not for a specific data set, but for the whole field at the macro level. Consequently, when considering the utilization of cultural features alone, compared with using other feature groups alone for prediction, cultural features can achieve the best overall performance, that is, our view has been successfully demonstrated.
>
> When cultural features are combined with other features, as shown in Table 2, it greatly improves the performance of the optimal transfer prediction. Specifically, **based on LangRank baseline, 9.97 MAP score and 8.30 NDGC score have been improved; based on MTVEC baseline, 25.00 MAP score and 11.38 NDGC score have been improved.** Observing Figure 4, for each dataset, even for the dataset of KOLD and Hindi, the model combined with cultural features can hit the GOLD top 1 transfer dataset  in the top3 prediction. Therefore, we are confident enough to argue that the cultural features have strong power.
>
> **R2. More details about the ZST process are needed.**
>
> A2. Thanks to the suggestion. We’ll add more details on the CR version.
>
> Specifically, for all 72 directed dataset pairs, we fine-tune the XLM-R [1] model on the transfer dataset with learning rate 1e-5. The training epoch is 20, training batch size is 8, we adopt the early-stopping method with patience 10 on the target dataset’s dev set. Macro F1 score is the metric.
>
> In order to mitigate the influence of randomness in our experiments and ensure result reliability, we run five times for each dataset pair. Subsequently, we compute the average performance (macro F1 score ) of the trained model on the target dataset's test set, representing the transfer performance for each dataset pair.
>
>
> **R3.The ground-truth ranking results generated from the ZST process with limited languages is based on a very strong assumption.**
>
> A3. Yes, you're right. We mention this in the Limitations section,  specifically in Line 582-585. However, it's worth noting that such assumptions are common in current research[2][3] to predict optimal transfer dataset or language. Since our novelty is focused on different aspects than this methodological one, we leave it to future work to generalize it.
>
>
>
> **Q1. I see that the authors are gonna perform experiments with pre-trained language models in future, but I'm curious why the authors did not do it in this paper**
>
> A4. Thanks to the reviewers for the suggestion to incorporate small pre-trained language models into the experiments. There is indeed an interest in this direction and it can be pursued even with limited computational resources at our disposal. However, we believe the gap to performing these experiments is still a conceptual one, as it requires a significant modification to the task setup, namely encoding the dataset contents and acquiring high-quality encoders for each target language and cultural context.
>
> Through elaborate experiments, we have shown that cultural features contribute to the prediction of optimal transfer dataset in transfer learning. Future work is expected to incorporate such cultural features directly into the pre-trained language model to enhance cross-cultural transfer learning strategies. Although we are still actively developing the exact methodology for achieving this objective, it represents an intriguing avenue for investigation and aligns with an inspiring research vision for follow-up work.
>
>
> References
> 1. https://huggingface.co/xlm-roberta-base
> 2. Choosing Transfer Languages for Cross-Lingual Learning (Lin et al., ACL 2019)
> 3. Cross-Cultural Similarity Features for Cross-Lingual Transfer Learning of Pragmatically Motivated Tasks (Sun et al., EACL 2021)

---

### Official Review · Reviewer_miiQ · 2023-08-05

**Typos Grammar Style And Presentation Improvements:** 1. when presenting offensive words, i…
**Soundness:** 4

**Ethical Concerns:**

Yes

**Excitement:**

3: Ambivalent: It has merits (e.g., it reports state-of-the-art results, the idea is nice), but there are key weaknesses (e.g., it describes incremental work), and it can significantly benefit from another round of revision. However, I won't object to accepting it if my co-reviewers champion it.

**Justification For Ethical Concerns:**

One of the most important contributions of this work is introducing a cultural dimension in predicting the ideal language to transfer from. In the experiments, authors use the cultural scores for each country (Figure 1, Table 8, and Table 9). I tried my best and cannot find the source of the values in the manuscript. A reliable source is important as these values may also encompass inferent biases towards certain cultures. Additionally, these values can be sensitive to time, thus requiring a documented time stamp.
Despite this, I do not rule out the possibility that I may have overlooked something.

**Paper Topic And Main Contributions:**

This paper investigates the feasibility of predicting the optimal source language for transfer learning on the offensive language detection task. The authors propose two factors -- cultural and offensiveness -- to assist in selecting optimal source language. They conduct sufficient experiments to show the effectiveness of the proposed factors.

**Questions For The Authors:**

A. The expression “predict language transfer” / “transfer prediction”  in the Introduction and the title of Sec 3 “How to predict transfer datasets” is hard to understand at first. As the objection of predicting is not specified by the expression. For example, predict the model’s performance after transferring to a different language? predict what aspect of the dataset?
Is it a conventional expression or it can be replaced with clearer phrases expressing “predict optimal language to transfer from”?

B. Where does the data for Fig. 1 come from?

C. Are the word embedding distances from Fasttext suppose to reflect the semantic difference or offensiveness distance of the words?
If it is the former, then two words with the same semantic meaning should be the same, if not, then it seems to be a limitation of the aligned Fasttext. If it is the latter, then how can Fasttext embedding reflect offensiveness?

D. Is there any evidence for arguments in Line 453-468?

**Reasons To Accept:**

The research question is important, and the proposed method is proven effective by experimental results.

**Reasons To Reject:**

Some arguments in the paper are not well-justified or accompanied by solid experimental results. For example, the source of cultural dimension values for each country is not specified.

**Reproducibility:**

3: Could reproduce the results with some difficulty. The settings of parameters are underspecified or subjectively determined; the training/evaluation data are not widely available.

**Reviewer Confidence:**

3: Pretty sure, but there's a chance I missed something. Although I have a good feel for this area in general, I did not carefully check the paper's details, e.g., the math, experimental design, or novelty.

---

> ### Author Rebuttal · Authors · 2023-08-28
>
> Thanks for your careful review. We hope that if our response is worthy enough, it will make you consider adjusting your score. We summarize your comments and respond in the following five aspects.
>
> **Q1. The source of cultural dimension values (from the rejected reason, question B and the ethical concern)**
>
> A1: These cultural dimension values are based on Hofstede’s Cultural Dimensions Theory, which is mentioned in Line 220-223.
>
> For more details, Hofstede’s Cultural Dimensions Theory serves as a framework utilized to comprehend cultural disparities among countries. A key component of this theory is the Hofstede Value Survey, a questionnaire designed to assess an individual's cultural values and beliefs. You can access additional information about the survey through the website https://geerthofstede.com/research-and-vsm/vsm-2013/  and retrieve the quantified cultural dimensions from https://geerthofstede.com/research-and-vsm/dimension-data-matrix/ **(The value used in Figure 1, Table 8, and Table 9 can be found in this website).**
>
> We appreciate your inquiry, and we’ll give these details in the CR version to avoid confusion.
>
> **Q2. Confusion about the expression such as “predict language transfer” / “transfer prediction”, and suggestion with a clearer phrase such as “predict optimal language to transfer from”**
>
> A2. Thanks for your question and sorry for the confusion. As you suggested, we can indeed add a descriptive term to define our prediction objective.Our aim in prediction is to identify the optimal (top) transfer datasets for the OLD task through the following steps: (1) train a set of NLP models with all available transfer datasets and collect evaluation scores, (2) train a ranking model to predict the ranking of transfer dataset, then obtain the top  transfer dataset.
>
> It's crucial to emphasize that our discussion involves transfer datasets rather than transfer languages, as languages do not inherently represent specific national cultures.  However, the dataset's specific sources and annotations can be specified to particular countries.
>
>
> **Q3.Question regarding the correlation between FastText embeddings and semantic distinctions, or the connection between FastText embeddings and the measure of offensiveness.**
>
> A3. These embeddings are usually described as semantic, and indeed, in this case we can say we are interested in the semantic distance between offensive words in the respective languages to quantify how similar their meaning is, and not what is the difference between their offensiveness.
>
>
> **Q4. Ask for evidence of arguments in Line 453-468?**
>
> A4. Khan [1] designed a study to identify the cultural differences between developed and developing countries, i.e., Australia and Pakistan, one representing western culture and the other representing eastern culture. Based on the cultural dimensions dissimilarities between these two cultures, the researcher developed the appended several propositions, including “High power distance employees will be less likely to perceive their supervisory abusive while” and “High individualistic cultural employees will be more likely to perceive their supervisory abusive”
>
> Bond et al.[2] conducted cross-cultural studies on college students from China and the United States. In the testing process, students should rate the legitimacy of the insult and the personality of the insulter after reading several scenarios. The researchers found that cultural variations (including Cultural Collectivism and Power Distance) in the use of social control are related to perceptions of an insult and of the insulter.
>
> Besides, this work[3] also highlights that people from different cultures exploit different categories of verbal abuse. In our analysis and experimentation (Subsection 5.3 and Table 5), we indeed observe that all cultural dimensions contribute to the accurate prediction of optimal transfer datasets for the OLD task.
>
> We will add these references to our CR version to enhance the soundness of our work.
>
>
> **Q5. Suggestion for presentation Improvements**
>
> A5. (1) Thanks for the nice suggestion about mask parts of offensive words with “*”. We will adopt this suggestion.
>
> In addition, we further adopt HARMCHECK [4] in our paper for handling and presenting harmful text in research. Our self-checks and measures are as follows:
>
>    1. **Risk of harm protocol:**  The specific risks of harm are derived from offensive language detection datasets in multiple languages. Experimenters for this work have access to these datasets, but they don't need to read them carefully.
>
>    2. **Preview:** The work will include a Content Warning directly after Abstract: “This document discusses examples of harmful content (hate, abuse, misinformation and negative stereotypes). The authors do not support the use of harmful language, nor any of the harmful representations quoted below.” The content warning is in red to maximize visibility.
>
>    3. **Distance:** Only a few offensive words are included to explain cross-cultural differences in the introduction section.
>
>    4. **Respect:** For slurs and profanities in text, the  authors will star out the first vowel with an asterisk (“*”)
>
>
> (2) Besides, we will revise the highlighting method for Table 3 based on the performance ranking of feature groups, aiming to enhance its readability.
>
>
> ### References
> 1. Impact of Hofstede’s Cultural Dimensions on Subordinate’s Perception of Abusive Supervision. (Khan, Shahid N., International Journal of Business and Management, 2014)
> 2. How are Responses to Verbal Insult Related to Cultural Collectivism and Power Distance? ( Michael H. Bond et al., Journal of Cross-Cultural Psychology, 1985)
> 3. Terms of abuse as expression and reinforcement of cultures. (Oudenhoven et al., International Journal of Intercultural Relations 2008)
> 4. Handling and Presenting Harmful Text in NLP Research (Kirk et al., Findings of EMNLP 2022)

---

### Official Review · Reviewer_CaTP · 2023-08-06

**Paper Topic And Main Contributions:** 1. This paper delves into the interse…
**Soundness:** 3

**Excitement:**

4: Strong: This paper deepens the understanding of some phenomenon or lowers the barriers to an existing research direction.

**Reasons To Accept:**

1. This is an interesting research finding. The authors validate that incorporating cultural features could improve the performance of cross-cultural transfer learning model.

2. This paper proposes six country-level features that quantify the variations in cultural values across different countries. It introduces a feature that measures the distance of offensive words between languages.

3. The logical structure of the paper is clear, and it revolves around the three research questions raised in the introduction in detail, and answers these three questions well.



**Reasons To Reject:**

1. For the problem definition in section 3, it should be more detailed to improve readability.

2. Symbols such as "P", "O", "C" in Table 2 have no detailed explanation.

3. The authors should compare to newer SOTA to show the effectiveness and generality of their features.


**Reproducibility:**

3: Could reproduce the results with some difficulty. The settings of parameters are underspecified or subjectively determined; the training/evaluation data are not widely available.

**Reviewer Confidence:**

4: Quite sure. I tried to check the important points carefully. It's unlikely, though conceivable, that I missed something that should affect my ratings.

---

> ### Author Rebuttal · Authors · 2023-08-28
>
> Thanks for your meticulous review of our work.  Based on your feedback, we will make revisions and additions to the manuscript for the CR version, aiming to enhance the soundness of our work. The response regarding the reasons for rejection is as follows:
>
> **R1&2: The definition in Section 3 needs to be more detailed, and the abbreviations in Table 2 require explanations.**
>
> A1&2: We will enhance the description in Section 3 to improve its readability, and mention the named abbreviation symbols in the caption of Table 2.
>
> **R3: The authors should compare to newer SOTA to show the effectiveness and generality of their features.**
>
> Thanks for the newer compared SOTA suggestion, which could enhance the soundness of our work.
>
> The domain research on optimal transfer language prediction and data (language) features is highly advanced and challenging. There are not many features that can be used for optimal transfer dataset prediction.
>
> After diligent exploration of relevant references, we are thrilled to have discovered a very recent and potentially baseline method named Colex2Lang[1]. The experimental results based on the baseline Colex2Lang are shown in Table (a) below. The experimental results show that our proposed features *CulDim* and *OffDist* can still enhance the newer baseline. This underscores the credibility and effectiveness of our introduced features.
>
> *Table (a) The performance of transfer language prediction on a newer baseline Colex2Lang. P, O and C stand for PRAG, OffDist, and CulDim respectively*
>
> |            | MAP       | NDGC      |
> |------------|-----------|-----------|
> | Colex2Lang | 57.78     | 71.41     |
> | +PRAG      | 55.77     | 71.95     |
> | +OffDist   | 59.64     | 73.81     |
> | +CulDim    | 73.25     | 71.56     |
> | +P+O       | 55.75     | 73.26     |
> | +P+C       | 69.63     | 76.33     |
> | +C+O      | **76.69** | **79.34**  |
> | +P+C+O     | 68.99     | 72.06     |
>
> ### References
> 1. Colex2Lang: Language Embeddings from Semantic Typology (Chen et al., NoDaLiDa 2023)

---

### Meta-Review · Area_Chair_afEL · 2023-09-23

**Recommendation:** 3

**Metareview:**

This paper looks at the problem of transfer learning for offensive language detection. While previous work has looked at language dimensions for finding an optimal source language, this work proposes looking at cultural differences. This paper tackles an important and timely problem with an interesting approach, so I believe this will contribute positively to the community.

There are a few improvements suggested by the reviewers. For one, clarity of writing could be improved. Other details are in the reviews, so please refer to those when making your revisions.

---

### Decision · Program_Chairs · 2023-10-07

**Decision:**

Accept-Findings

**Comment:**

This paper looks at the problem of transfer learning for offensive language detection. While previous work has looked at language dimensions for finding an optimal source language, this work proposes looking at cultural differences. This paper tackles an important and timely problem with an interesting approach, so I believe this will contribute positively to the community.

There are a few improvements suggested by the reviewers. For one, clarity of writing could be improved. Other details are in the reviews, so please refer to those when making your revisions.